

# Genome-wide identification and expression analysis of C2H2 zinc finger proteins in Chinese jujube (*Ziziphus jujuba* Mill.) in different fruit development stages and under different levels of water stress

Xie Zhengwan[1], Ji Qing[1], Lihu Wang[2], Ao Zhang[1], Shengxing Li[3], Sunyang Li[1], Mei Chen[1], Yang Jiayue[1] and Ruifang Wang[1]

[1] Puer University, Puer, China
[2] School of Landscape and Ecological Engineering, Hebei University of Engineering, Handan, China
[3] Camphor Engineering Technology Research Center for National Forestry and Grassland Administration, Jiangxi Academy of Forestry, Nanchang, China

Corresponding authors
Lihu Wang, hdwanglh@163.com
Ruifang Wang,
peurdr_wang@foxmail.com

## ABSTRACT

The C2H2 type zinc finger proteins (C2H2-ZFPs) are prevalent motifs found widely across the eukaryotic kingdom, playing crucial roles in various biological processes, including stress responses and plant growth and development. To date, C2H2-ZFPs have been identified in many plant systems, but there have been no reports in Chinese jujube (*Ziziphus jujuba* Mill.). In this study, a total of 77 ZjC2H2-ZFPs were identified in Chinese jujube and classified into three groups, with set A containing 29 genes, set B containing two genes, and set C containing 46 genes. The set A group genes were further divided into three groups: A1, A2, and A4 (with no member of the A3 subgroup found in jujube). The set C C2H2-ZFPs genes were also further divided into three groups: C1 containing 24 genes, C2 with two genes, and C3 with three genes. These ZjC2H2-ZFPs were distributed on 12 chromosomes and one tandem duplicated pair of ZjC2H2-ZFPs was found on chromosome 4 (ZjC2H2-21 with ZjC2H2-22). Notably, the 77 ZjC2H2-ZFPs identified in this study lacked finger clusters comprising 10 or more repeats. The structure and protein motif analysis of ZjC2H2-ZFPs showed that most C1 subgroup members were enriched with 'QALGGH' motif zinc finger helices and the A1a ZjC2H2-ZFPs contained highly conserved 'SATALLQKAAQMGS' residues in jujube. A unique finding was the discovery of a conserved non-finger domain (PCYCC motif) in A1 group members, absent in other ZjC2H2-ZFPs and unreported in other species. The enzyme activity of jujube leaves under different water stress treatments were measured, and the results showed that as the degree of water stress increased, the activity of SOD enzymes and $H_2O_2$ content also increased. The POD enzyme activity levels of different treatment groups were CK>MS>SS>LS. The levels of malondialdehyde (MDA) content observed under various treatments were notably higher and the proline content was lower in comparison to the control group (CK). Differential expression of ZjC2H2-ZFPs and specific responses were analyzed under water stress and different fruit development stages of jujube using RNA-Seq data. The correlation between expression patterns and protective enzyme activities

under water stress was also examined. The results indicated that the expression levels of different ZjC2H2-ZFPs varied. A further protein interaction analysis indicated that ZjC2H2-ZFPs serve as pivotal transcriptional regulators with diverse functions, encompassing DNA or RNA binding and participation in protein interactions, with ZjC2H2-20, ZjC2H2-36, and ZjC2H2-57 being potential key players in these regulatory processes. Their roles appear particularly crucial in responding to abiotic stresses like water stress and regulating plant hormones. This study provides valuable insights into understanding stress responses and enhancing the quality of Chinese jujube during breeding.

## INTRODUCTION

Zinc finger proteins (ZFPs) are comprised of one or more zinc-finger domains (ZF). Within a ZF domain, the coordination of cysteine (C) and/or histidine (H) residues conforms to the three-dimensional structure characteristic of the finger type (*Arrey-Salas et al., 2021*; *Liu et al., 2015*). ZFPs participate in the regulation of important biological processes unique to plants. The C2H2-type zinc finger proteins (C2H2-ZFPs) represent one of the prevalent motifs among zinc finger proteins (ZFPs) and are widely distributed across the eukaryotic kingdom. These C2H2-ZFPs consist of one or more zinc-finger domains (ZF). As a transcription factor, the C2H2-type zinc finger motif can be represented by the sequence Cys-X2-4-Cys-X12-His-X3-5-His, with X indicating any amino acid residue, which forms the conservative DNA-binding motif of C2H2-ZFPs (*Englbrecht, Schoof & Böhm, 2004*; *Yin et al., 2020*). The C2H2-ZFPs are divided into three subgroups (set A, B, and C) according to the structure and distribution of their finger domains. Sets A and B comprise proteins with tandemly arranged ZF domains, the C2H2-ZFPs of set B contain TF3A transcription factor, and set C comprises proteins with a single isolated finger or multiple dispersed fingers (*Englbrecht, Schoof & Böhm, 2004*; *Mathieu et al., 2003*). C2H2-ZFPs are involved in many biological processes, encompassing pivotal roles in plant development, growth, and the intricate orchestration of stress responses. Their multifaceted functions have garnered extensive scholarly attention, with in-depth investigations conducted across a diverse spectrum of organisms (*Alam et al., 2019*; *Han et al., 2020*; *Liu et al., 2017*; *Wang et al., 2019a*). The expressions of C2H2-ZFPs in tomato responded to drought, heat, cold, salt, and pathogenic stresses (*Hu et al., 2019*; *Zhao et al., 2020*). A previous investigation into poplar revealed that C2H2 Q-type genes responded to abiotic stresses. These genes were found to be triggered by hydrogen peroxide and the phytohormone jasmonate, and were involved in the regulation of various signaling molecules (*Gourcilleau et al., 2011*). The expression of GmZFP3 with the conserved QALGGH motif of soybean C2H2-type ZFP responded to the induction of the abscisic acid and hormone polyethylene glycol, and transgenic *Arabidopsis* experiments showed that *GmZFP3* negatively regulated drought responses (*Zhang et al., 2016*). The *StZFP1* of the potato C2H2-ZFP gene is a typical
TFIIIA-type two-finger zinc finger gene, which responded to abiotic and biotic stress, with expression increasing after salt stress (*Tian et al., 2010*).

Chinese jujube (*Ziziphus jujuba* Mill.), known colloquially as red date or Chinese date, is a botanical specimen belonging to the *Rhamnaceae* family. It is one of the earliest domesticated fruit-bearing trees, and is thus one of the most significant large fruit species on a global scale (*Ji et al., 2023*; *Liu et al., 2020a*). There has been a recent increase in acknowledgment of the economic, ecological, and societal importance of jujube trees. The cultivation of jujube trees has also extended to numerous countries and regions across the globe (*Wang et al., 2019b*; *Wang et al., 2019c*).

C2H2-type zinc finger genes have been extensively studied in various plant systems (*Agarwal et al., 2007*; *Alam et al., 2019*; *Jiao et al., 2020*; *Liu et al., 2015*; *Liu et al., 2020b*; *Wang et al., 2019a*; *Yin et al., 2020*); however, there have been no prior reports on their identification in Chinese jujube. Furthermore, the expression patterns of *ZjC2H2-ZFPs* in response to water stress and during different developmental stages of Chinese jujube remain largely unexplored. This study conducted an extensive analysis of C2H2-ZFPs in Chinese jujube, including gene structure, conserved motifs, physicochemical properties of the encoded proteins, exon/intron composition, chromosomal positioning, expression patterns, and their potential correlations with protective enzyme activities. These findings not only contribute to a comprehensive characterization of *ZjC2H2-ZFPs,* but also establish a solid foundation for future functional analyses of *ZjC2H2-ZFPs* in Chinese jujube.

## MATERIALS AND METHODS

### Identification of the C2H2-ZFP gene family in Chinese jujube

The genomic and protein sequence data for Chinese jujube were acquired from the Jujube Genome Database (https://www.ncbi.nlm.nih.gov/assembly/GCF_000826755.1; *Liu et al., 2014*). The HMM profiles related to ZFP domains, which encompass the zinc finger protein (ZFP) domain (PF00096), were acquired from the Pfam website (http://pfam.xfam.org/; *Finn et al., 2016*). TBtools (*Chen et al., 2020*) was used to identify all C2H2-ZFP genes based on the HMM profiles of these domains from Pfam. The Modular Architecture Research Tool (SMART, http://smart.embl-heidelberg.de/) and the NCBI Conserved Domain Database (CDD, https://www.ncbi.nlm.nih.gov/cdd; *Marchler-Bauer et al., 2015*) were used to validate the presence of at least one ZFP domain in all C2H2-ZFP genes (*Letunic et al., 2004*). Any genes lacking ZFP domains were eliminated from the pool of candidates. As reported by Englbrecht, Schoof, and Böhm in 2004, the 176 sequences of zinc finger proteins were obtained from Arabidopsis thaliana. These sequences were sourced from the Arabidopsis Information Resource (TAIR; http://www.Arabidopsis.org/; *Englbrecht, Schoof & Böhm, 2004*).

### Chromosomal location, gene structure, motif, and protein characterization analysis

The gene positions of ZjC2H2-ZFPs on Chinese jujube chromosomes were obtained from the GFF3 annotation file. A nomenclature update for these genes was then performed, considering their locations and the linear arrangement relative to their respective

chromosomes. TBtools (*Chen et al., 2020*) was then used to identify the gene structure of Chinese jujube C2H2-ZFP genes. The exon/intron organization of ZFPs was visualized by aligning their coding sequences (CDS) with their corresponding genomic sequences with the help of TBtools (*Chen et al., 2020*). The Multiple Expectation Maximization for Motif Elicitation (MEME) program (version 5.0.5; *Bailey et al., 2009*) was used to identify conserved motifs within the *ZjC2H2-ZFPs* with the following parameter settings: maximum number of motifs to find, 10; minimum and maximum width of motifs, 6 and 50, respectively; and default parameters. The tools provided by the ExPASy Server (https://prosite.expasy.org/; *Gasteiger et al., 2005*) were used for protein characterization and analysis. These tools helped predict various attributes of *ZjC2H2-ZFPs*, including their length, amino acid composition, theoretical isoelectric point, instability index, molecular weight (MW), and atomic composition.

## Multiple sequences alignment and phylogenetic analysis

The complete protein sequences of C2H2-ZFPs from both *Arabidopsis thaliana* and Chinese jujube were used for the phylogenetic analysis. Multiple sequence alignments were performed using the MUSCLE tool (https://www.ebi.ac.uk/jdispatcher/msa/muscle?stype=protein). A phylogenetic tree was then inferred using IQTREE with a bootstrap value of 1000 (*Nguyen et al., 2015*) for statistical support. To enhance the presentation of the phylogenetic tree and provide annotations, TBtools (*Chen et al., 2020*) was used for visualization and annotation.

## *ZjC2H2-ZFPs* expression profile analyses

To investigate the role of *ZjC2H2-ZFPs* in Chinese jujube during various fruit development stages and in response to water stress, an analysis of the expression profiles of each *ZjC2H2-ZFPs* was conducted using previously generated RNA-seq transcriptomic data. These RNA-seq datasets were obtained using an Illumina HiSeq 2000 and comprise data from different jujube developmental stages. The data was stored in the NCBI Sequence Read Archive and can be accessed using the accession number SRP162927 (*Qing et al., 2019*). Additionally, the researchers collected RNA-seq transcriptomic data for jujube leaves subjected to various water stress conditions, although this dataset has been uploaded China National GeneBank DataBase (sub050826). A heatmap illustrating the expression profiles of the C2H2-ZFP genes in jujube was created based on fragments per kilobase of exon per million mapped reads (FPKM) values calculated from RNA-Seq reads. This heatmap was created with the Tbtools software (*Chen et al., 2020*).

## Protein-protein interaction network prediction

A protein-protein interaction analysis was conducted using the protein sequences of *ZjC2H2-ZFPs* as query sequences on the STRING website (https://string-db.org/; *Szklarczyk et al., 2023*) with a stringent confidence threshold set at an option value greater than 0.7. To establish meaningful comparisons, orthologs were selected from *Arabidopsis thaliana* as the reference species. Following the BLAST step, which was performed with default settings, an interaction network among *ZjC2H2-ZFPs* was constructed where the gene with the highest score (bitscore) was considered a key factor.

## Plant materials and water stress

Annual trees of the Chinese jujube cultivar, "Dongzao", were selected as the plant materials for this study. The plants were potted and then exposed to one of four different water stress treatments: mild stress (LS) controlled at 60%–70% of the maximum field water capacity, moderate stress (MS) controlled at 40%–45% of the maximum field water capacity, severe stress (SS) controlled at 25%–30% of maximum field water capacity, or control (CK) at 75%–80% of the maximum field water capacity. Based on previous research results and experience (*Cruz et al., 2012*; *Tian, 2014*; *Xiuping et al., 2002*), it was determined that the experimental stress time would be 20 days to prevent leaf damage and plant death. The weighing method and a soil moisture content analyzer (Horde) were used to measure soil moisture content. Each plant received an initial watering and then were not watered until the soil moisture content naturally dried to the set soil moisture content range. The initial total weight was then measured and the surface of the basin was covered with plastic film to prevent water evaporation. The weighing method was then used to replenish water daily. Each treatment was conducted with three biological replicates.

After subjecting the plants to 20 days of stress treatment, three-to-five mature leaves were carefully selected between 8:00 and 9:00 in the morning, and then promptly frozen using liquid nitrogen and stored in a low-temperature refrigerator at −80 °C. The leaf sample was bifurcated into two portions: one portion was allocated for transcriptome analysis, while the other part was designated for assessing physiological and biochemical parameters. This assessment encompassed a range of indicators using specialized kits (Solarbio, Beijing, China), including determining superoxide dismutase (SOD) activity, hydrogen peroxide ($H_2O_2$) content, peroxidase (POD) activity, malondialdehyde (MDA) content, catalase (CAT) activity, and proline content. According to the manufacturer's instructions, 0.2 g samples were ground into powder in liquid nitrogen and the extraction solution was added. This was then centrifuged for 10 min at 8000 rpm at 4 °C, then the supernatants were used for measuring SOD, POD, MDA, CAT, and $H_2O_2$ using corresponding assays. After adding corresponding reagent, the activity of SOD, POD, CAT, and $H_2O_2$ were calculated at 450 nm, 470 nm, 240 nm, and 415 nm, respectively. For MDA content, the absorbance at 600 nm, 532 nm, and 450 nm was measured and the MDA content was calculated using the difference in absorbance between 532 nm and both 450 nm and 600 nm. The proline content was measured with the acidic-ninhydrin method at 520 nm (*Bates, Waldren & Teare, 1973*).

## Correlation analysis

The correlation analysis of *ZjC2H2-ZFPs* expression with enzyme activity indexes was performed using the original FPKM values. The "corr.test" function of the R software was used for the Pearson correlation analysis, and significance testing and plotting were performed using R's "coroplot" software package.

## Total RNA extraction and qRT-PCR analysis

The methods for extracting total RNA from jujube leaves in this study were adopted from previously published studies, specifically following the procedures outlined in *Gao et al.*

*(2021)*. The gene expression and transcriptome data were validated with qRT-PCR assays. The primer sequences used for these experiments were designed using Primer Premier 5.0 software (*Lalitha, 2000*) and can be found in Table S1. Quantitative RT-PCR was performed using the Fast Super EvaGreen qPCR Master Mix (US Everbright Inc., Suzhou, China) and Rotor-Gene Qreal-time PCR system instrument (Qiagen, Hilden, Germany). The *ZjActing* (107425564; *Liu et al., 2019*) jujube gene served as the internal reference gene and changes in gene expression were calculated using the $2^{-\Delta\Delta CT}$ method (*Livak & Schmittgen, 2001*).

### *ZjC2H2-ZFPs* synteny and cis-element analysis

This study used the MCScanX tool to perform a homology analysis of the *ZjC2H2-ZFPs* between jujube and two other plants, *Arabidopsi* and peach (*Wang et al., 2012*), and used TBtools (*Chen et al., 2020*) to construct a syntenic analysis map. The sequence information of *Arabidopsis* and peach were downloaded from the TAIR database (http://www.arabidopsis.org/; *Swarbreck et al., 2007*) and Genome Database for Rosaceae (GDR; https://phytozome.jgi.doe.gov/; *The International Peach Genome Initiative et al., 2013*; *Jung et al., 2014*; *Verde et al., 2017*). TBtools was used to obtain the cis-element of *ZjC2H2-ZFPs*, spanning 1500 base pairs upstream from the translation initiation site (*Chen et al., 2020*). The acquired promoter sequences underwent analysis using the "Search for CARE" tool within PlantCARE (http://bioinformatics.psb.ugent.be/webtools/plantcare/html/). The results were visualized using TBtools (*Chen et al., 2020*).

## RESULTS

### Identification of jujube ZjC2H2-ZFPs and analysis of their protein physicochemical properties

In total, 77 proteins encompassing one or more ZF (zinc finger) domains were identified in the jujube genome database (Tables S2 and S3). The C2H2 genes were named *ZjC2H2-1* to *ZjC2H2-77* according to their location on the jujube chromosome. The physiological and biochemical information of the 77 *ZjC2H2-ZFPs* were analyzed (Table S2). The lengths of the *ZjC2H2-ZFPs* ranged from 174 (*ZjC2H2-52*) to 1,454 (*ZjC2H2-27*) amino acids. The *ZjC2H2-ZFPs* identified in jujube displayed variations in molecular weight, ranging from 19,064.1 (*ZjC2H2-52*) to 162,471.58 (*ZjC2H2-27*) kilodaltons (kDa), and in theoretical isoelectric point, ranging from 4.57 (*ZjC2H2-7*) to 9.63 (*ZjC2H2-66*), with an average theoretical isoelectric point of 7.86. The instability index of *ZjC2H2-ZFPs* was greater than 30, and the value of GRAVY (grand average of hydropathy) was negative, which showed that the *ZjC2H2-ZFPs* were unstable hydrophilic proteins.

### General classification and characterization of *ZjC2H2-ZFPs*

This study referred to the classification method of *Arabidopsis thaliana* and classified the *ZjC2H2-ZFPs* based on the tandem arrays of fingers of proteins (*Wolfe, Nekludova & Pabo, 2000*). In this study, 77 proteins were identified and classified. As shown in Fig. 1 and Table S3, 29 *ZjC2H2-ZFPs* were in set A, two *ZjC2H2-ZFPs* were in set B, and 46 *ZjC2H2-ZFPs* were in set C. A total of 31 *ZjC2H2-ZFPs* contained tandem ZF arrays, with 29 *ZjC2H2-ZFPs* in set A containing three-to-five ZFs and two *ZjC2H2-ZFPs* (TF3A) in set

B containing up to nine ZFs. Notably, the nine fingers of one of the *Zj-TF3As* (*ZjC2H2-74*) was arranged in the same way as that of *Arabidopsis thaliana* TF3A (*Englbrecht, Schoof & Böhm, 2004*), and they were all arranged differently from animal TF3A (*Mathieu et al., 2003*). A distance of 1–10 amino acids between two fingers is a tandem arrangement/short linker, and a distance of more than 10 amino acids is a dispersed arrangement/long spacer (*Englbrecht, Schoof & Böhm, 2004*). In *ZjC2H2-74* and *At-TF3A*, the initial finger is noticeably distant from the rest, while fingers 2–4 and 5–9 are organized into two distinct tandem arrays. Based on finger characteristics, set A was divided into three subgroups, namely A1, A2, and A4, which differed from *Arabidopsis thaliana* as no member of A3 subgroup was found in jujube, while there was one A3 subgroup member found in *Arabidopsis thaliana*. The A1 subgroup was classified into four different subsets named A1a, A1b, A1c, and A1d with 13, one, three, and seven members, respectively (Fig. 1 and Table 1).

Set C (46 members) was the largest group in the *ZjC2H2-ZFPs* (about 60%) and comprised proteins with one isolated finger or two-to-five dispersed fingers. According to the characteristics of set C members, set C was further classified into three subgroups based on the number of amino acids between the last two Histidine (H) of each finger: C1 with three amino acids, C2 with four, and C3 with five. The C1 subgroup was the largest with 36 members, followed by the C2 subgroup with nine members, and the C3 subgroup with only one member. Most members of the C1 subgroup contained QALGGH motif zinc finger helices, which is consistent with *Arabidopsis thaliana* and petunia (*Englbrecht, Schoof & Böhm, 2004*; *Kubo et al., 1998*).

According to number of fingers, the C1 subgroup was further divided into C1-1, C1-2, C1-3, C1-4, and C1-5 with 19, 11, one, two, and one members, respectively. In addition, referring to the classification of *Arabidopsis thaliana*, two *ZjC2H2-ZFPs* (*ZjC2H2-6* and *ZjC2H2-63*) with C1 and C2 finger characteristics were put into the C1 subgroup and classified as a C1C2mixed subset.

### *ZjC2H2-ZFPs* structure and protein motif analysis

The gene exon/intron structures were analyzed in the *77 ZjC2H2-ZFPs* (Fig. 2). The CDSs were disrupted by introns in all groups except groups C1C2mixed and C1-5. The A1a group was disrupted by 3.1 introns, on average. The A1b group had only one member and was interrupted by two introns. Among the three members of group A1c, *ZjC2H2-36* contained one intron and the other two members had no introns. In group A1, *ZjC2H2-58* contained two introns, while all other group members were disrupted by one intron. In group A2, one member was disrupted by six introns and the other was disrupted by two introns. The A4 group was disrupted by 4.3 introns, on average. The two members of group B were disrupted by seven and six introns, respectively. Most members of group C1 had no introns and only one exon. The A4 group was disrupted by 3.7 introns on average. These results indicate that alterations in the exon/intron structure of *ZjC2H2-ZFPs* may have significant implications for their functional variations throughout the course of evolution.
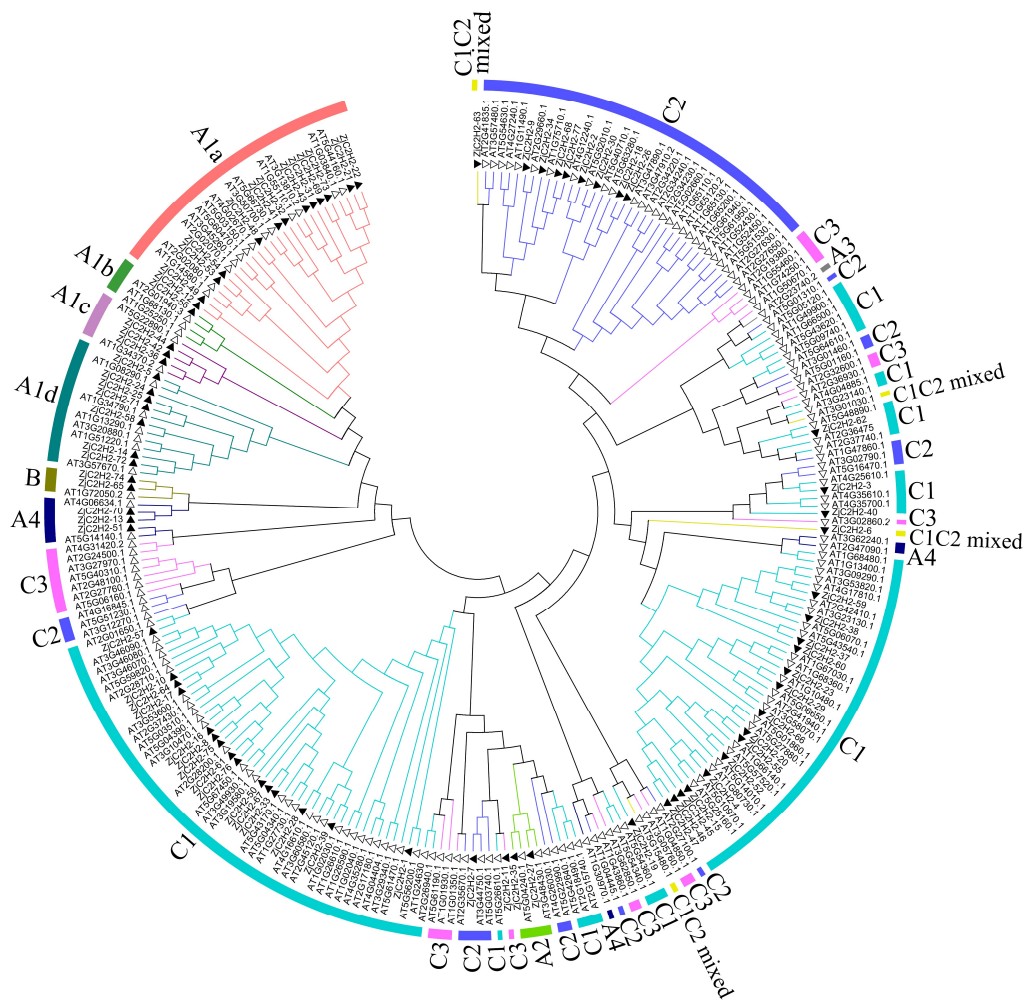

**Figure 1** **Phylogenetic tree of the C2H2 zinc finger protein of jujube and Arabidopsis.** The small white triangles represent the ZjC2H2-ZFPs of jujube and the small black triangles represent the AT-ZFPs of Arabidopsis.

**Table 1** **The zinc finger domain distribution characteristic and other conserved motifs of groups A1a to A1d.**

| Group | Finger 1 | Finger 2 | Finger 3 | Finger 4 | Other domain |
|-------|----------|----------|----------|----------|--------------|
| A1a | **CX2CX12HX3H** | *CX4CX17HX4H* | CX2CX12HX3C | *CX1CX12HX3C* | **SATALL** |
| A1b | **CX2CX12HX3H** | *CX4CX17HX4H* | CX2CX12HX3C | *CX1CX12HX3C* | |
| A1c | **CX2CX12HX3H** | **CX4CX20HX4H** | **CX2CX13HX3C** | **CX1CX12HX6H** | |
| A1d | **CX2CX12HX3H** | **CX4CX20HX4H** | CX2CX12HX3C | CX1CX12HX7H | PCYCC |

The MEME program successfully forecasted ten conserved motifs among the *ZjC2H2-ZFPs* found in jujube (Fig. 3). All *ZjC2H2-ZFPs* contained motif 1, which can be represented by the sequence CX2CX12HX3H. Table 1 illustrates the distribution pattern of zinc finger

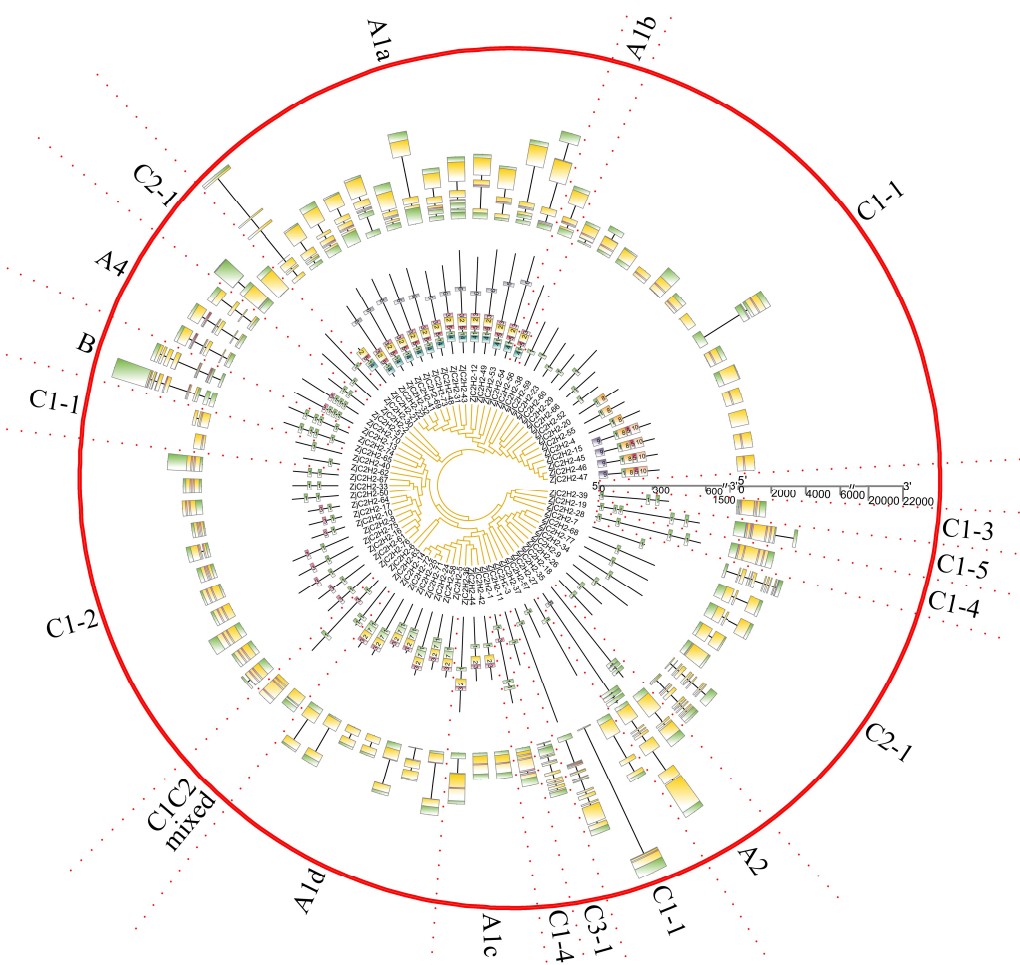

**Figure 2 The gene structure and conserved motifs analysis of ZjC2H2-ZFPs of jujube.**

domains and other conserved motifs within groups A1a to A1d. Except for *ZjC2H2-21*, which contained motifs 4, 1, 5, and 2, all other members of the A1a group contained motifs 4, 1, 5, 2, 3, and 6. Motif 1 and motif 3 constituted the first and fourth zinc finger domain of the A1a group, respectively. The latter part of motif 5 and the front part of motif 2 constituted the second finger domain in jujube. In addition, the four finger domain distribution of A1b and A1a had the same characteristics, but A1a contained the other conserved sequence at the C-terminus, as shown in motif 6 as 'SATALLQKAAQMGS', which was consistent with that in *Arabidopsis thaliana* (*Englbrecht, Schoof & Böhm, 2004*). The A1c group contained motifs 1, 2, and 3. The A1d group had one more conservative motif than A1c, which was motif 7. The first and second finger domain distributions of A1c and A1d had the same characteristics. Notably, the sequence logos of the motif 7 repeats (Fig. 3) showed A1d *ZjC2H2-ZFPs* contained the 'PCYCC' conserved sequence, which was not found in other *ZjC2H2-ZFPs*. This pattern may play a crucial role in facilitating interactions or determining the localization of *ZjC2H2-ZFPs*. The members of groups A2, A4, and B contained highly conserved zinc finger 1 domains. Of the two members of

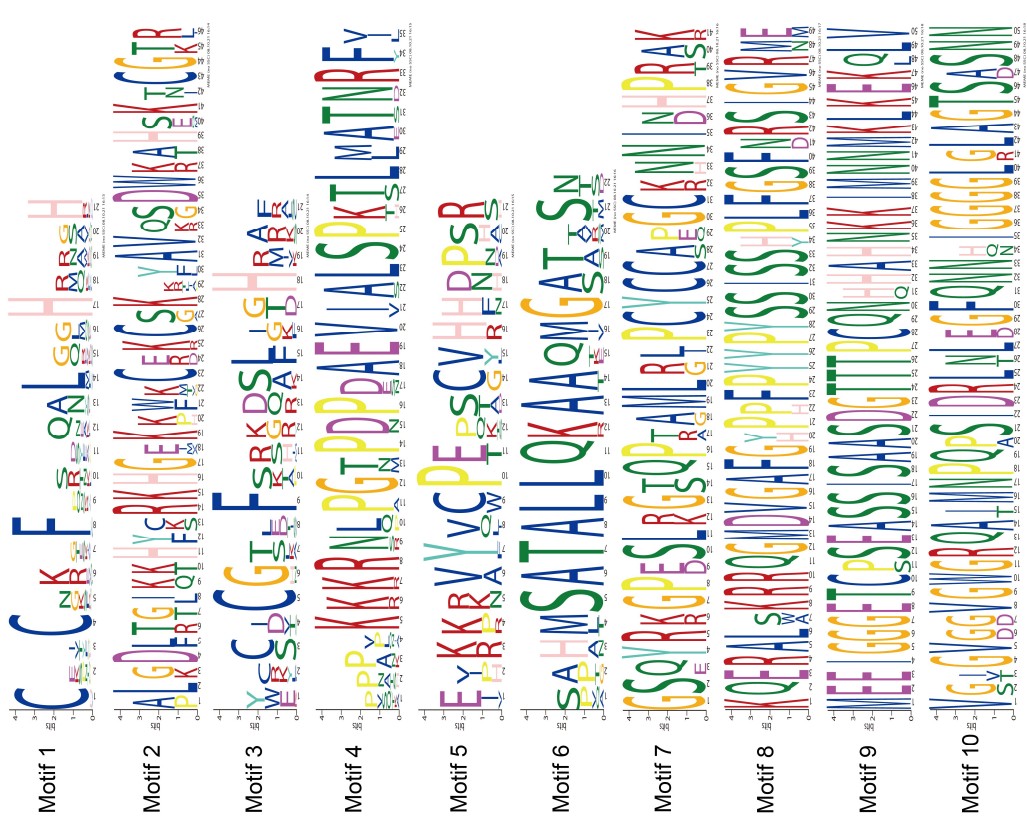

**Figure 3    The conserved motifs sequences of the ZjC2H2-ZFPs.**

A2, *ZjC2H2-35* contained three motif 1 repeats and *ZjC2H2-27* contained three motif 1 repeats and one motif 6. Among the three members of A4, *ZjC2H2-51* contained two motif 1 repeats and *ZjC2H2-13* and *ZjC2H2-70* contained three motif 1 repeats and one motif 3. Both members of group B contained four motif 1 repeats. The members of subgroup C also contained highly conserved zinc finger 1 domains. Twelve of the 19 members of group C1-1 only contained one motif 1, three members contained motifs 1 and 8, and the other four members contained motifs 9, 1, 8, 5, and 10. Three of the 11 members in group C1-2 contained two motif 1 repeats, while the other eight members contained motifs 1 and 3. Groups C1-3, C1-4, C1-5, C2, and C3 contained three, four, four, one, and one highly conserved motif 1, respectively.

## Distribution on the chromosome and tandem duplicated analysis of *ZjC2H2-ZFPs*

The 77 *ZjC2H2-ZFPs* within the jujube genome exhibited an uneven distribution across chromosomes *Zj01* to *Zj12* (Fig. 4), with chromosome 2 containing the most *ZjC2H2-ZFPs* (10) and chromosome 11 containing the fewest, with just one *ZjC2H2-ZFP*. A gene duplication analysis revealed that one tandem duplicated pair of *ZjC2H2-ZFPs* was found on chromosome 4 (*ZjC2H2-21* with *ZjC2H2-22*), which differs from cucumber as there were no segmental duplication events observed in the *CsZFPs* of cucumber (*Yin et al.,*

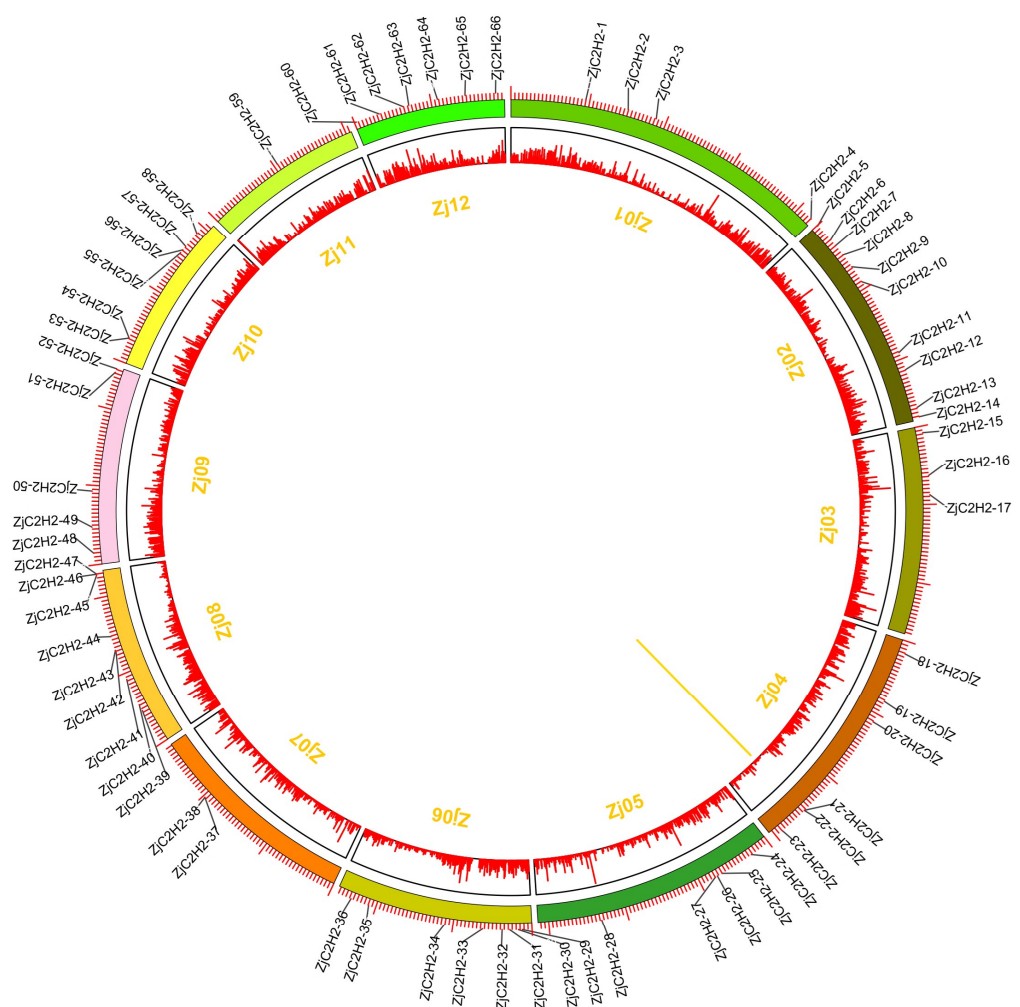

**Figure 4  Chromosomal distribution and gene duplication of ZjC2H2-ZFPs of jujube.** The inner ring denotes the pseudochromosomes of jujube. The outer ring shows the chromosomal distribution of jujube ZjC2H2-ZFPs, and the scale is five Mb. The yellow line inside the circle are the tandem duplicated ZjC2H2-ZFP pairs.

*2020*). However, the number of tandem repeat gene pairs in jujube was less than that in *Arabidopsis thaliana* (*Englbrecht, Schoof & Böhm, 2004*), rice (*Agarwal et al., 2007*), and *Medicago truncatula* (*Jiao et al., 2020*).

## Interaction network of *ZjC2H2-ZFPs* in Chinese jujube

In order to pinpoint potential associations among *ZjC2H2-ZFPs* and their interacting proteins and protein complexes, prediction networks were established using STRING 18. These networks were built upon the foundation of the interaction network involving orthologous genes in *Arabidopsis thaliana*, as depicted in Fig. 5. Finally, 23 identified *Arabidopsis thaliana* proteins were predicted to participate in the interaction network, which corresponded to 35 *ZjC2H2* homologous genes of jujube. In the *At-TF3A* homologous gene (*ZjC2H2-74* and *ZjC2H2-65*) associated network, this gene interacted directly

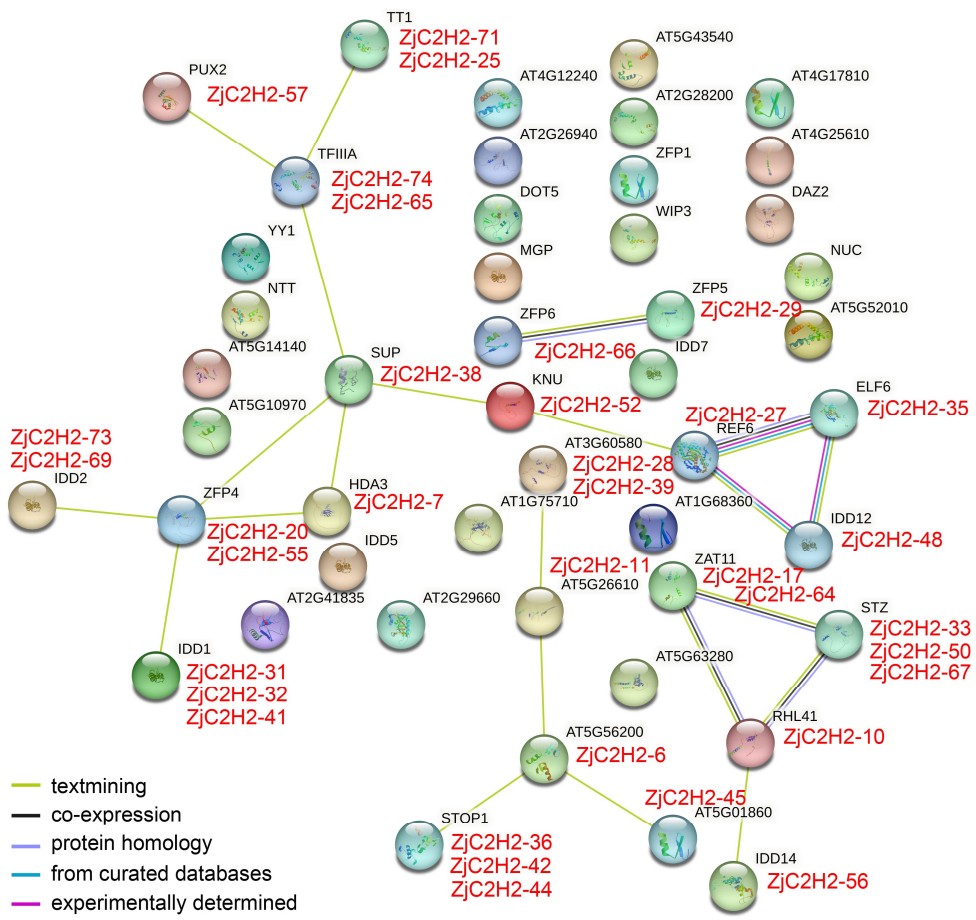

**Figure 5 The network of ZjC2H2-ZFPs of jujube functional connections.** Colored balls in the network were used as a visual aid to indicate different input proteins and predicted interactors.

with four *ZjC2H2-ZFPs*: *ZjC2H2-71*, *ZjC2H2-25*, *ZjC2H2-57*, and *ZjC2H2-38*. The *AtZFP4* homologous gene (*ZjC2H2-20* and *ZjC2H2-55*) interacted directly with seven *ZjC2H2-ZFPs*: *ZjC2H2-38*, *ZjC2H2-7*, *ZjC2H2-31*, *ZjC2H2-32*, *ZjC2H2-41*, *ZjC2H2-73*, and *ZjC2H2-69*. The *AtREF6* homologous gene (*ZjC2H2-27*) interacted directly with three *ZjC2H2-ZFPs*: *ZjC2H2-52*, *ZjC2H2-35*, and *ZjC2H2-48*. The *AtRHL41* homologous gene (*ZjC2H2-10*) interacted directly with six *ZjC2H2-ZFPs*: *ZjC2H2-17*, *ZjC2H2-64*, *ZjC2H2-33*, *ZjC2H2-50*, *ZjC2H2-67*, and *ZjC2H2-56*. The *ZjC2H2-6* as *AT5G56200* homologous gene interacted directly with five *ZjC2H2-ZFPs*: *ZjC2H2-11*, *ZjC2H2-45*, *ZjC2H2-36*, *ZjC2H2-42*, and *ZjC2H2-44*. The *ZjC2H2-38*, as the *AtSUP* homologous gene, interacted directly with six *ZjC2H2-ZFPs*: *ZjC2H2-74*, *ZjC2H2-65*, *ZjC2H2-52*, *ZjC2H2-7*, *ZjC2H2-20*, and *ZjC2H2-55*. The *ZjC2H2-52* as *AtKNU* homologous gene interacted directly with two *ZjC2H2-ZFPs*: *ZjC2H2-38* and *ZjC2H2-27*. The *ZjC2H2-66*, as the *AtZFP6* homologous gene, interacted directly with *ZjC2H2-29*, which was the *AtZFP5* homologous gene.

The identified gene functions in the interaction results of STRING indicated that of the 23 identified proteins of *Arabidopsis thaliana*, the *AtZAT11*, *AtSTZ*, *AT3G60580*, and
*AtREF6* proteins might be involved in variety of stress responses. *AtREF6* and *AtHDA3* proteins are involved in transcriptional regulation. *AtZFP4*, *AtZFP5*, *AtZFP6*, *AtIDD1*, and *AtIDD14* are involved in the regulation of plant hormones, *AtELF6* is involved in brassinosteroid signaling, and *AtTT1* is involved in the regulation of pigment synthesis and flavonoid accumulation. *AtTT1*, *AtSUP,* and *AtKNU* proteins are involved in regulating cell differentiation. *AtPUX2* might mediate the powdery mildew-plant interaction. *AtSTOP1* and *AtRHL41* might be involved in abiotic stress resistance. Further research is needed to elucidate the role of *ZjC2H2-ZFPs* in responding to stress signaling and regulating plant development in jujube.

## Analysis of *ZjC2H2-ZFPs* expression profiling from RNA-Seq data

To detect the role of *ZjC2H2-ZFPs* in different developmental stages of jujube and in response to abiotic stress, the expression levels of each *ZjC2H2-ZFP* were analyzed through the available RNA-seq transcription data obtained by the previous research group. According to the log2-transformed FPKM values of the dataset, all expressing *ZjC2H2-ZFPs* (with an FPKM value larger than 1) were collected and the different expression patterns of the *ZjC2H2-ZFPs* are shown in Fig. 6. Based on the expression profiles of the *ZjC2H2-ZFPs*, the expression levels of *ZjC2H2-ZFPs* in different fruit development stages and under different water stresses of jujube were divided into seven subgroups: a1-a7 (Fig. 6A) and b1-b7 (Fig. 6B), respectively. In Figs. 6A, Y, EN, WM, HR, and FR represent the young fruit stage, the fruit enlargement stage, white mature fruit stage, half-red fruit stage, and the full-red fruit stage of jujube, respectively. The accuracy and reliability of the RNA-Seq data under various water stress conditions were additionally confirmed by qPCR experiments. These experiments were conducted specifically for eight chosen *ZjC2H2-ZFPs* with different expression trends across four distinct treatment conditions (Fig. 6C).

Members of *ZjC2H2-ZFPs* in the a1 subgroup were highly expressed in the early stages of fruit development (Y and EN stages). The a2 subgroup members were highly expressed at the Y, EN, and WM stages of jujube. The a3 subgroup members were lowly expressed at the later stages of fruit development (HR and FR stages). Conversely, the a5 and a6 subgroup members had higher expression levels in the later fruit stages. The members of subgroup a7 were the most highly expressed at fruit maturity (FR stage). All of these findings collectively indicate that the differential expression patterns observed in *ZjC2H2-ZFPs* likely hold significant and diverse functional implications during various developmental stages of jujube.

In response to water stress, the members of subgroups b2, b3, and b4 were highly induced. In particular, the members of subgroup b2 showed high expression after being highly induced by LS treatment, and then the expression decreased with the increase of water stress. The members of subgroup b7 showed a change pattern of down-regulation and then up-regulation in response to water stress. Moreover, in response to water stress, most of the *ZjC2H2-ZFPs* showed increased or decreased expression patterns in LS, MS, and SS treatment of jujube leaf, indicating these *ZjC2H2-ZFPs* function differently in response to water stress. These results indicate that progressively more significant roles are
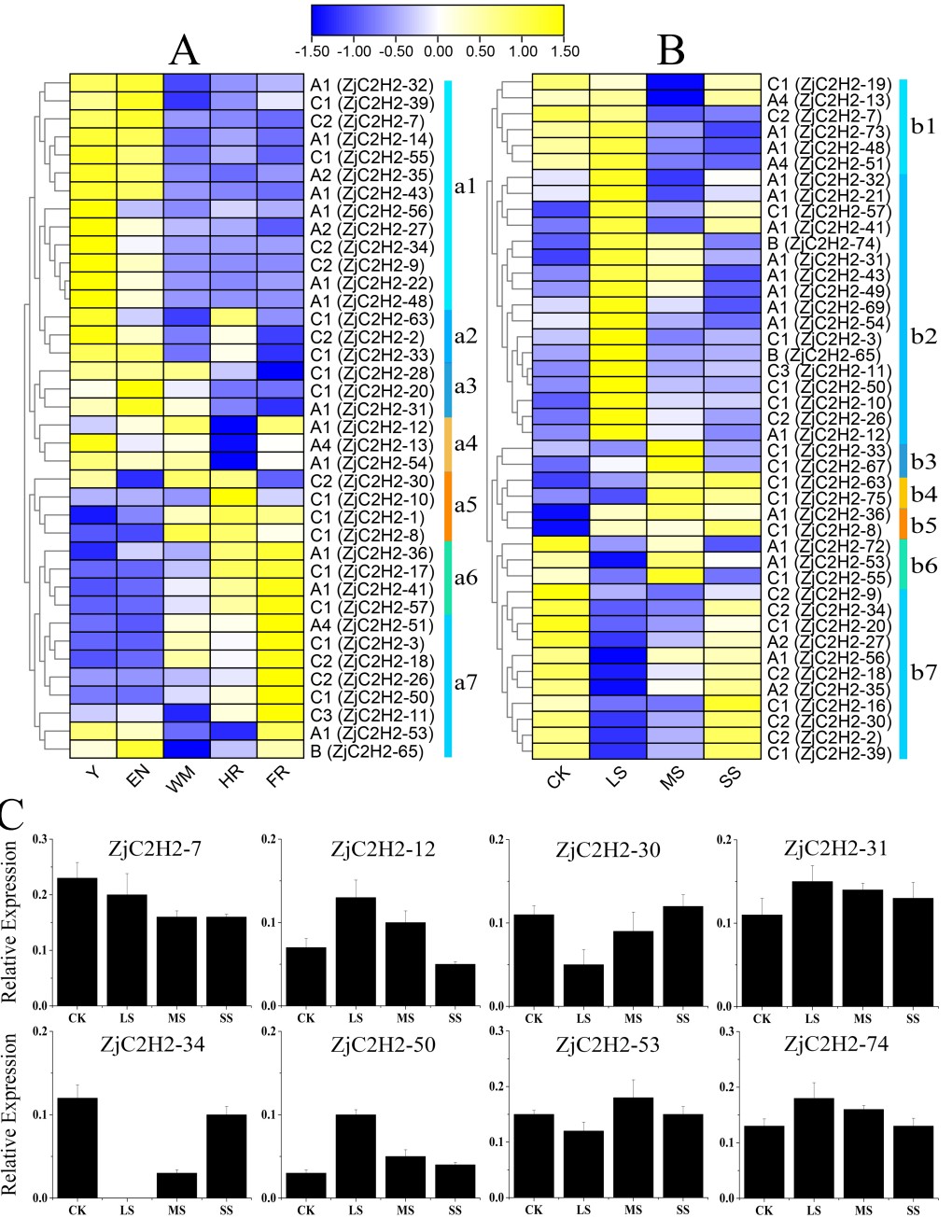

**Figure 6   Heatmap of the expression levels of ZjC2H2-ZFPs and the expression levels of 8 ZjC2H2-ZFPs by qRT-PCR.** (A) Expression levels of ZjC2H2-ZFPs in different development stages of jujube. (B) Expression levels of ZjC2H2-ZFPs under different water stress. (C) Expression analysis of 8 ZjC2H2-ZFPs in jujube by qRT-PCR. The data were normalized to the ZjActin gene.

played by *ZjC2H2-ZFPs* during plant growth and development, and in response to abiotic stresses in the jujube plant.

**Table 2  Effect of water stress on protective enzyme activities in jujube leaves.**

| Treatment | SOD (U/g) | POD (U/g) | CAT (U/g) | MDA (nmol/g) | $H_2O_2$ ($\mu$mol/g) | Proline (g/kg) |
|---|---|---|---|---|---|---|
| CK | 630.85 ± 19.01d | 86931.97 ± 685.16a | 1385.92 ± 10.92a | 59.94 ± 1.06c | 5.73 ± 0.24d | 0.175 ± 0.0029a |
| LS | 794.60 ± 68.88c | 55909.43 ± 678.79d | 1255.08 ± 27.67b | 71.87 ± 1.05a | 7.55 ± 0.31c | 0.045 ± 0.0017b |
| MS | 914.52 ± 42.69b | 75926.12 ± 804.25b | 1423.33 ± 52.94a | 70.05 ± 1.73ab | 8.41 ± 0.22b | 0.035 ± 0.001c |
| SS | 1021.59 ± 32.08a | 68229.46 ± 498.09c | 997.16 ± 74.93c | 68.91 ± 0.43b | 9.62 ± 0.51a | 0.044 ± 0.0012b |

## Protective enzyme activities determination and correlation analysis

The enzyme activity of jujube leaves under different water stress treatments were measured (Table 2), and the results showed that as the degree of water stress increased, the activity of SOD enzymes and the contents of $H_2O_2$ showed an upward trend. There were significant differences in SOD enzyme activity, $H_2O_2$ content, and POD enzyme activity among different comparison groups. The POD enzyme activity levels of different treatment groups were CK>MS>SS>LS. Under MS stress treatment, the peak CAT enzyme activity recorded reached 1,385.92 U/g, which was not a significant difference compared to the control group (CK), but was significantly different than the LS and SS treatment groups. The levels of MDA content observed under water stress treatments were notably elevated compared to the control group (CK), increasing by 19.90%, 16.87%, and 14.96% in the LS, MS, and SS groups compared to the CK control group, respectively. The proline content exhibited a substantial reduction in all treatment groups compared to the control group, with decreases of 74.29%, 80.00%, and 74.86% recorded in the LS, MS, and SS groups, respectively.

To further explore the connections between the expression patterns of the 43 ZjC2H2-ZFPs and the activities of protective enzymes, a correlation analysis was performed using R software (Fig. 7). The results showed that SOD activity was positively correlated with $H_2O_2$ content and the *ZjC2H2-63* gene. There was an inverse correlation observed between POD activity and the *ZjC2H2-57* gene. Additionally, MDA content was negatively correlated with both proline content and the *ZjC2H2-20* and *ZjC2H2-9* genes. Proline content was positively correlated with *ZjC2H2-9* and negatively correlated with *ZjC2H2-36*.

### *ZjC2H2-ZFPs* synteny and cis-element analysis

A total of 39 pairs of paralogous *ZjC2H2-ZFPs* in jujube and peach (Fig. 8; Table S4) and 20 pairs of orthologous *ZjC2H2-ZFPs* in jujube and *Arabidopsis* were identified (Fig. 8; Table S4). A total of 35 *ZjC2H2-ZFPs* showed homology with peach and 17 showed homology with *Arabidopsis.* Of those 17 *ZjC2H2-ZFPs,* all of them except *ZjC2H2-63* shared homology with peach (Table S4). An in-depth analysis of cis-acting elements within members of the *ZjC2H2* gene family revealed a total of 12 distinct types across the 77 identified *ZjC2H2* genes (Fig. 9). The distribution of cis-acting element types varied from one to five within each family gene member. Notably, genes such as ZjC2H2-1, *ZjC2H2-19, ZjC2H2-25, ZjC2H2-49,* and *ZjC2H2-59* exclusively exhibited a single type of cis-acting element, encompassing elements related to light response, modules for light response, and elements involved in both abscisic acid and light responses. *ZjC2H2-44* stood out by harboring the most diverse range of cis-acting element types, including

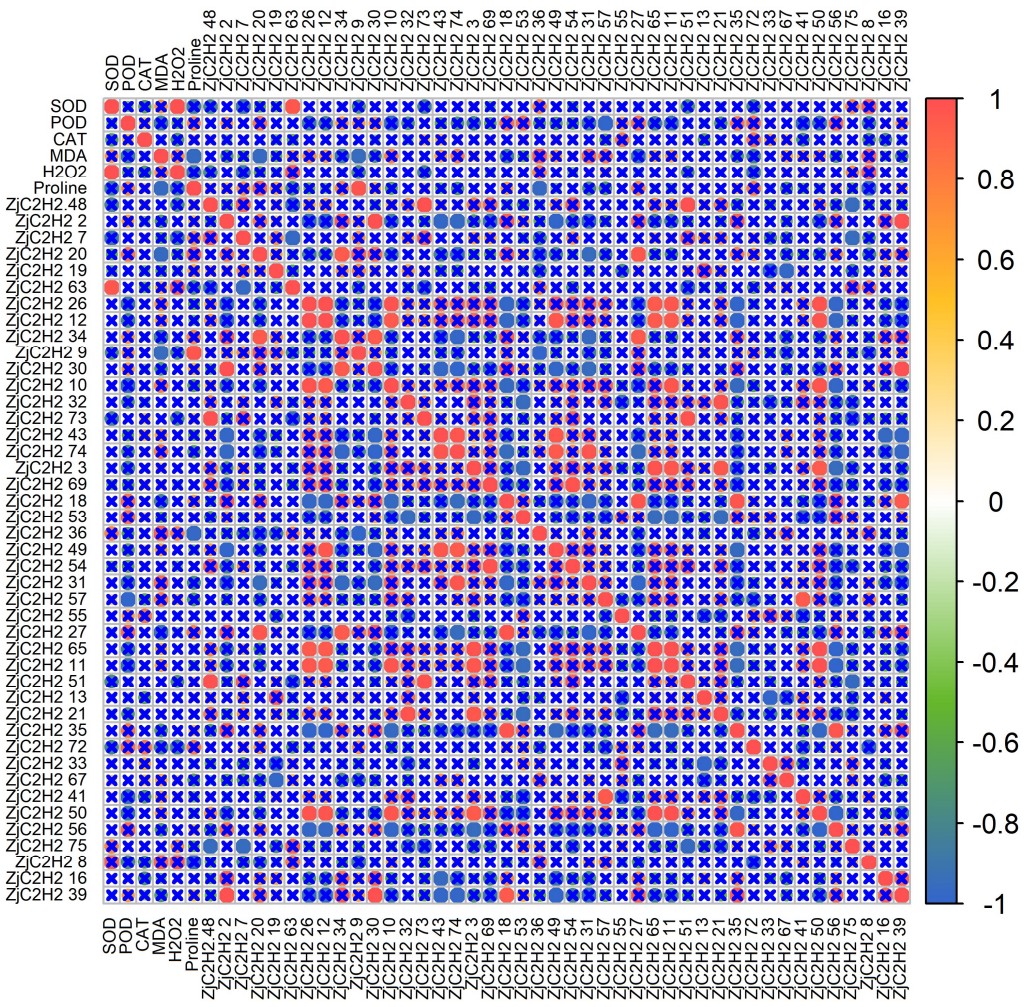

**Figure 7  The correlation analysis of protective enzyme activities with ZjC2H2-ZFPs expression in ju-jube leaf.** The correlation analysis was performed using the original FPKM counts and the expression levels of the 43 ZjC2H2-ZFPs and the protective enzyme activities under four different water stress treatments of jujube leaf. The "×" symbol indicates that the correlation is not significant at the 0.05 level. Red and blue circles represent a significant positive correlation and a significant negative correlation at the 0.05 level, respectively.

homeopathic elements, encompassing part of a light response element, cis-acting elements involved in abscisic acid response, light response elements, cis-acting regulatory elements related to meristem expression, MYB binding sites associated with drowsy index, and additional parts of light response elements. Among the identified cis-acting elements, the highest prevalence was observed for those involved in abscisic acid response, with a total of 93 instances distributed across 47 *ZjC2H2* genes. These results indicate that this gene family may also play a significant role in responding to the biochemical processes related to light.

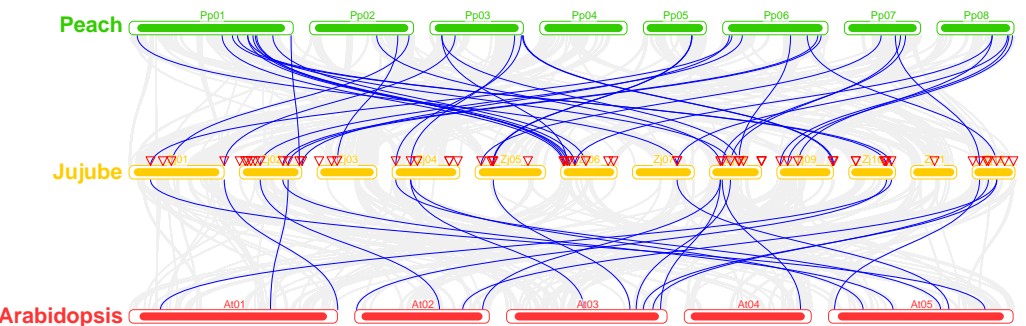

**Figure 8** Syntenic relationships of C2H2-ZFPs among Peach, Jujube and Arabidopsis.

## DISCUSSION

The C2H2-ZFP gene family is a significant transcription factor family in plants, playing a crucial role in regulating various aspects of plant growth, development, and responses to stressors. While this family has been extensively studied in various plant species, it has not yet been explored in Chinese jujube, a unique plant with economic, medicinal, and ecological significance. This study identified 77 C2H2-ZFP genes in the Chinese jujube genome, which is fewer than have been found in other plants like *Arabidopsis* (176; *Englbrecht, Schoof & Böhm, 2004*), *Oryza sativa* (189; *Agarwal et al., 2007*), *Medicago truncatula* (218; *Jiao et al., 2020*), *Zea mays* (211; *Wei, Pan & Li, 2016*), *Glycine max* (321; *Yuan et al., 2018*), *Populus* (109; *Liu et al., 2015*) and *Solanum lycopersicum* (99; *Zhao et al., 2020*), but more than those found in yeast *Böhm, Frishman & Mewes, 1997*. These differences are likely due to variations in the evolutionary paths of these plants. In addition, sixteen *ZjC2H2-ZFPs* of jujube showed close homology to the genes of peach and *Arabidopsis*, indicating that these 16 ZjC2H2-ZFPs, as well as their orthologs, likely existed before the ancestral divergence of the C2H2-ZFPs.

A gene duplication analysis identified a tandem duplicated pair of ZjC2H2-ZFPs on chromosome 4 (ZjC2H2-21 with ZjC2H2-22), a pattern also observed in *Arabidopsis thaliana* (*Englbrecht, Schoof & Böhm, 2004*), rice (*Agarwal et al., 2007*), and *Medicago truncatula* (*Jiao et al., 2020*), but not in cucumber (*Yin et al., 2020*). Tandem duplicates refer to adjacent paralogous genes on the same chromosome. Similar duplication events were observed in the jujube MYB superfamily, indicating recent genome-wide duplications in the jujube genome (*Cannon et al., 2004*; *Qing et al., 2019*).

Among the 77 *ZjC2H2-ZFPs*, none contained finger clusters with 10 or more repeats (*Böhm, Frishman & Mewes, 1997*). Instead, they were divided into sets, with set A having tandem arrays of three-to-five finger domains, in contrast to *Arabidopsis*, which has two set A members with only two fingers (*Englbrecht, Schoof & Böhm, 2004*).

Analyzing the physicochemical attributes, structural features, and protein motifs provided insights into the evolutionary variations within each ZjC2H2-ZFP member. Within the C1 subgroup, encompassing 36 members and previously denoted as the EPF-family in Petunia (*Kubo et al., 1998*), it is noteworthy that most of its members

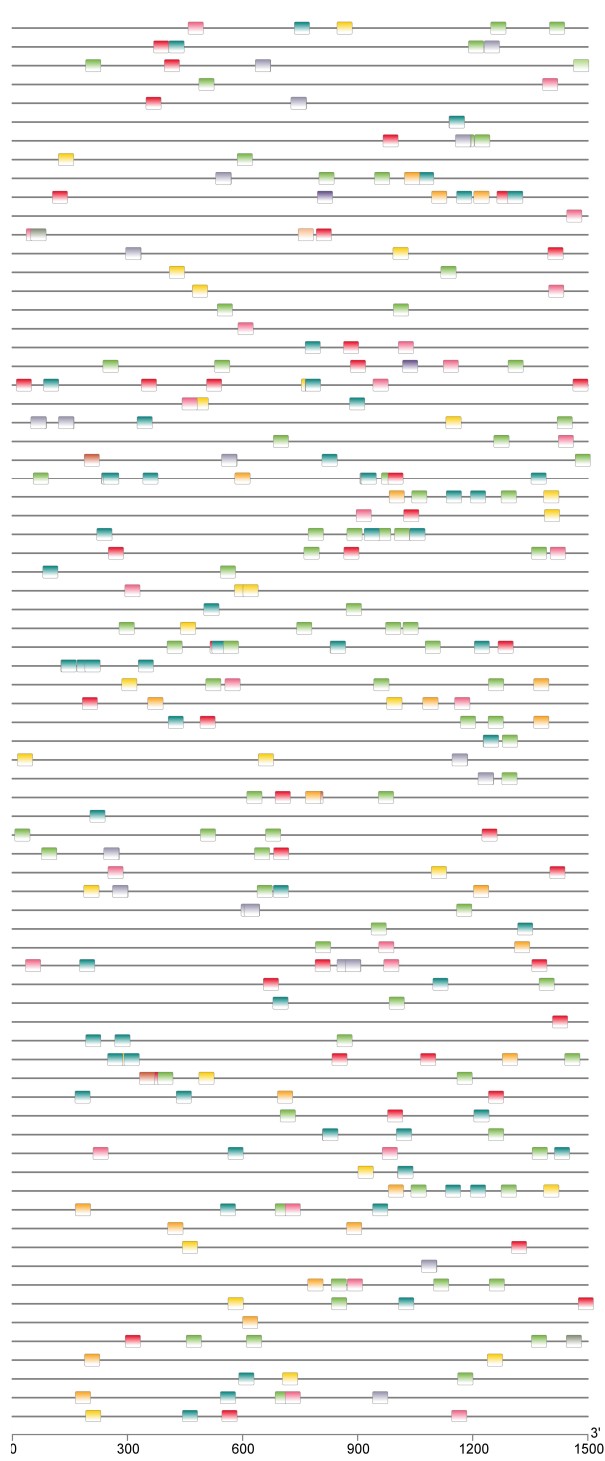

part of a light responsive element
MYB binding site involved in drought-inducibility
part of a module for light response
cis-acting element involved in the abscisic acid responsiv
light responsive element
cis-acting regulatory element involved in light responsive
part of gapA in (gapA-CMA1) involved with light responsi
cis-acting regulatory element related to meristem expres
element involved in differentiation of the palisade mesop
part of a conserved DNA module involved in light respon
cis-acting regulatory element related to meristem specific
protein binding site

**Figure 9   Cis-regulatory element analysis of ZjC2H2-ZFPs.**

possessed 'QALGGH' motif zinc finger helices. This distinctive motif is also prevalent in *Arabidopsis thaliana* and petunia, implying that these C1 subgroup members may play roles in plant-specific life processes (*Kubo et al., 1998*).

Both sets A and B consisted of proteins featuring tandemly arranged ZF (zinc finger) domains. However, a conserved non-finger domain was also identified, characterized by the PCYCC motif, within the members of the A1d group (Table 1 and Fig. 3), which was not found in other *ZjC2H2-ZFPs* and has not been reported in other species. This pattern may be essential for the interactions or localization of C2H2-ZFPs. One notable example from set B was the *Zj-TF3A* protein, specifically *ZjC2H2-74*, which possesses nine ZF domains organized in more than one array. Remarkably, the first ZF finger was isolated, while fingers 2–4 and 5–9 are structured into two distinct tandem arrays. This arrangement mirrors a similar configuration found in *Arabidopsis*, but differs from the structure of the TF3A protein in animals (*Englbrecht, Schoof & Böhm, 2004*). At-TF3A has been successfully cloned and characterized. This protein exhibits specific binding affinity to 5S rDNA and 5S rRNA, and is capable of enhancing the transcription of a 5S rRNA gene in *in vitro* experiments (*Mathieu et al., 2003*). In this study, *ZjC2H2-74* as the TF3A homologous gene, may have the same function, as it interacted directly with four *ZjC2H2-ZFPs* (*ZjC2H2-71*, *ZjC2H2-25*, *ZjC2H2-57,* and *ZjC2H2-38*) in the protein interaction network analysis. The *ZjC2H2-57* gene was negatively correlated with POD activity in the water stress treatment studies of jujube. These results suggest that *ZjC2H2-ZFPs* assume crucial roles as pivotal transcriptional regulators, with roles ranging from binding to DNA or RNA to participating in protein interactions under abiotic stress conditions, which aligns with prior research findings (*Englbrecht, Schoof & Böhm, 2004*; *Mishra et al., 2014*; *Yin et al., 2020*).

This study measured enzyme activity under water stress and correlated it with gene expression. The results showed that *ZjC2H2* genes (including *ZjC2H2-9*, *ZjC2H2-20*, *ZjC2H2-36*, *ZjC2H2-57*, and *ZjC2H2-63*) were significantly associated with various enzyme activities like SOD, POD, MDA, and proline content. Additionally, *ZjC2H2-20*, *ZjC2H2-36*, and *ZjC2H2-57* were identified as homologous genes to AtZFP4, AtSTOP1, and AtPUX2 proteins in *Arabidopsis thaliana*. These genes are known to respond to adversity stress, participate in hormonal regulation, and engage in gene interactions. This aligns with the predictions made in cucumber and *Arabidopsis* studies regarding the important roles of C2H2-ZFPs in responding to various stresses and phytohormone regulation (*Ciftci-Yilmaz & Mittler, 2008*; *Lu & Huang, 2008*; *Yin et al., 2020*).

## CONCLUSIONS

In this study, a total of 77 *ZjC2H2-ZFPs* were identified in Chinese jujube, and classified into three groups, with 29, two, and 46 *ZjC2H2-ZFPs* belonging to sets A, B, and C, respectively. These *ZjC2H2-ZFPs* were distributed on 12 chromosomes and one tandem duplicated pair of *ZjC2H2-ZFPs* was found on chromosome 4 (*ZjC2H2-21* with *ZjC2H2-22*).

An analysis of ZjC2H2-ZFP protein properties revealed their nature as unstable hydrophilic proteins. Similar to higher eukaryotes, the 77 ZjC2H2-ZFPs lack finger clusters

with 10 or more repeats. The nine fingers of Zj-TF3A (ZjC2H2-74), a homologous gene to At-TF3A, align in the same way as *Arabidopsis thaliana*. A conserved non-finger domain (PCYCC motif) was identified in A1d group members, which was not observed in other ZjC2H2-ZFPs and has not been reported in other species. It is speculated that this pattern may be necessary for interactions or localization of C2H2-ZFPs. Sixteen *ZjC2H2-ZFPs* of jujube showed close homology to the genes of peach and *Arabidopsis*, indicating that these 16 *ZjC2H2-ZFPs,* as well as their orthologs, likely existed before the ancestral divergence of the C2H2-ZFPs. Enzyme activity assays under different water stress treatments revealed increased SOD activity and H2O2 levels as water stress levels increased. POD enzyme activity levels under different levels of water stress were CK>MS>SS>LS. MDA content increased and proline content decreased under various water stress treatments. An expression analysis revealed that *ZjC2H2-ZFPs* responded to water stress and exhibited stage-specific expression during jujube development. This study suggests that *ZjC2H2-ZFPs* likely play crucial roles as primary transcriptional regulators, engaging in diverse activities such as DNA or RNA binding and active participation in protein interactions. Notably, *ZjC2H2-20*, *ZjC2H2-36*, and *ZjC2H2-57* may have significant regulatory functions in stress response and plant hormone regulation. These findings offer new perspectives on stress responses and quality improvement in Chinese jujube breeding.

### Funding

This study was funded by the Yunnan Fundamental Research Projects (grant NO. 202101AU070030), the National Natural Science Foundation of China (grant number 31870277), the Hebei Natural Science Foundation (C2022402021), the Science and Technology Research Project of University in Hebei Province (QN2020205), the Science and Technology Research and Development Plan Project of Handan (19422011008–49), the Education Talent Cultivation Project of Puer University (2023JYRC0004). The funders had no role in study design, data collection and analysis, decision to publish, or preparation of the manuscript.

### Grant Disclosures

The following grant information was disclosed by the authors:
Yunnan Fundamental Research Projects: 202101AU070030.
The National Natural Science Foundation of China: 31870277.
Hebei Natural Science Foundation: C2022402021.
The Science and Technology Research Project of University in Hebei Province: QN2020205.
Science and Technology Research and Development Plan Project of Handan: 19422011008–49.
Education Talent Cultivation Project of Puer University: 2023JYRC0004.

### Competing Interests

The authors declare there are no competing interests.

## Author Contributions

- Xie Zhengwan performed the experiments, prepared figures and/or tables, authored or reviewed drafts of the article, and approved the final draft.
- Ji Qing conceived and designed the experiments, performed the experiments, analyzed the data, prepared figures and/or tables, authored or reviewed drafts of the article, and approved the final draft.
- Lihu Wang conceived and designed the experiments, performed the experiments, analyzed the data, authored or reviewed drafts of the article, and approved the final draft.
- Ao Zhang analyzed the data, prepared figures and/or tables, and approved the final draft.
- Shengxing Li analyzed the data, authored or reviewed drafts of the article, and approved the final draft.
- Sunyang Li analyzed the data, prepared figures and/or tables, and approved the final draft.
- Mei Chen performed the experiments, authored or reviewed drafts of the article, and approved the final draft.
- Yang Jiayue analyzed the data, prepared figures and/or tables, and approved the final draft.
- Ruifang Wang conceived and designed the experiments, authored or reviewed drafts of the article, and approved the final draft.

## Data Availability

Raw data are available in the Supplemental Files.

## Supplemental Information

Supplemental information for this article can be found online at http://dx.doi.org/10.7717/peerj.18455#supplemental-information.

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
