# Peer review of "Genome-wide identification and expression analysis of C2H2 zinc finger proteins in Chinese jujube (Ziziphus jujuba Mill.) in different fruit development stages and under different levels of water stress"

_PeerJ, doi:10.7717/peerj.18455_

## Round 0.1 · original submission · Major Revisions

Please revise the article considering the reviewers comments.

Reviewer 2 has requested that you cite specific references. You may add them if you believe they are especially relevant. However, I do not expect you to include these citations, and if you do not include them, this will not influence my decision.

**Language Note:** The review process has identified that the English language must be improved. PeerJ can provide language editing services - please contact us at [email protected] for pricing (be sure to provide your manuscript number and title). Alternatively, you should make your own arrangements to improve the language quality and provide details in your response letter. – PeerJ Staff

Reviewer 1 ·

Basic reporting

1. The manuscript is interesting and will be useful for Jujube breeding. The introduction part was well written. However, the abstract was found just general information but not written properly and failed to convey the importance and finding of the manuscript. It was not mentioned properly what are the differential expression for water stress and developmental stages. There were grammatical errors in the manuscript and Arabidopsis thaliana may be italic in manuscript.

Experimental design

The materials and methods was not properly written. These are many lacuna in this part. Some of the major suggestions were provided as follow.
1. In the manuscript, Dongzao cultivars were used for water stress treatment. The treatment method was not elaborated in the materials and method of the manuscript. What is the age of the plant for water stress treatment. How the different field water capacity was maintained for 20 days during the stress treatment?
2. Why water stress treatment for 20 days was given? Any specific reasons, may be included in the manuscript.
2. Multiple Expectation Maximization for Motif Elicitation (MEME) reference may be provided.
3. RNA-seq transcriptomic data for jujube 159 leaves subjected to various water stress conditions has not been submitted to NCBI or SRA. Authors are encouraged to submit the RNA-seq data to NCBI or public database before article publication.
4. Why the experimental methods of physiological and biochemicals parameters such as superoxide dismutase (SOD) activity, hydrogen peroxide (H2O2) content, peroxidase (POD) 186 activity, malondialdehyde (MDA) content, catalase (CAT) activity and proline content was not explained in the manuscript?
5. References for software used in the manuscript should be included.
6. How qRT-PCR assays was conducted? what is ZjACT used as internal control reference gene?

Validity of the findings

The materials and methods was not scientifically written. How the nomenclature of the 77 ZjC2H2-ZFPs identified in the present manuscript were given? This may be explained in the manuscript.

·

Basic reporting

In the following manuscript, the authors performed only basic bioinformatics analysis and expression analysis, etc, however, it is strongly recommended to perform, Domain figure, cis-regulatory element analysis, miRNA prediction, Ka/Ks, synteny analysis, collinearity analysis with other genomes, 3D modeling and TFs network analysis, etc, before acceptance and publications.

Experimental design

In the following manuscript, the authors performed only basic bioinformatics analysis and expression analysis, etc, however, it is strongly recommended to perform, Domain figure, cis-regulatory element analysis, miRNA prediction, Ka/Ks, synteny analysis, collinearity analysis with other genomes, 3D modeling and TFs network analysis, etc, before acceptance and publications.

Validity of the findings

In the following manuscript, the authors performed only basic bioinformatics analysis and expression analysis, etc, however, it is strongly recommended to perform, Domain figure, cis-regulatory element analysis, miRNA prediction, Ka/Ks, synteny analysis, collinearity analysis with other genomes, 3D modeling and TFs network analysis, etc, before acceptance and publications.

Additional comments

In the current manuscript, authors Ji Qing et al presented the research work entitled “Genome-wide identification and expression analysis of C2H2 zinc finger proteins in Chinese jujube under fruit development and water stresses”. In this study, a systematic investigation of the C2H2 gene family in Jujube was performed with multiple bioinformatics and expression analyses. A total of 77 ZjC2H2 genes were identified from the Jujube genome. Moreover, Phylogenetic, structural analysis, motifs, and domain analysis were performed. Differential ZjC2H2-ZFPs expression and specific responses were analyzed under water stresses of jujube based on RNA-Seq data, and the correlation between expression patterns and protective enzyme activities underwater stresses were analyzed. This work provides a comprehensive understanding of the C2H2 gene family and valuable clues for future studies on the function and evolution of C2H2 genes in Jujube. Overall, the manuscript is in the area of interest but there are still revisions that need to be addressed.
Line 2: Title: write the botanical name of (Chinese Jujube).
Line 42: Replace “To data “ with “To date”.
It is recommended to use the botanical names of all the mentioned plants, such as line 43 “Chinese Jujube”.
Line 50: Italicized “ZjC2H2-ZFPs”.
Line 50: Replace “Our” with “current study”.
The author performed qPCR but did not mention the results in the abstract, therefore it is recommended to add the qPCR expression results also and provide the more significant C2H2 genes for further functional characterizations.

The abstract is not well written, such as the author has performed “protective enzyme activities” but there is no result explanation in the abstract in this part, it is recommended to mention all the results in the abstract.

It is recommended to use the botanical names of all the mentioned plants, provide the genome version and database links used for genome downloading, and also provide the protein sequences used for identifications and phylogenetic analysis in supplementary.

In addition, the author used “We” and “Our” words more often in this manuscript which is not the scientific way of writing, it is suggested to do corrections accordingly.

Line 167: Italicized “Arabidopsis thaliana”.

Line 173-178: How does the author say that the water stress capacity is 25 %, - 75 %? Has the author used any instrument to measure the water capacity? Explain it

Line 196: In Table S1 the author mentioned 10 pairs of genes including ZjACT, while in results Figure 6 C only 8 genes qPCR results, recheck and do corrections.
Line 196: How did the author select genes for qPCR? Mention it in material/results.
Line 215 -217: These lines are meaningless, recheck and do corrections.
The discussion and conclusion are too long, rewrite in a concise style.

The genome-wide gene family is popular with a multi and advanced bioinformatics analysis, However, in the current manuscript, the author only performed the basic analysis and no any advanced analysis such as collinearity, TF networking, miRNAs etc, therefor It is recommended to check carefully entire manuscript, perform the remaining analysis, and add the following recent publications as references in the introduction, analysis, and discussion :
Rizwan, et al. "Genome-wide identification and expression profiling of KCS gene family in passion fruit (Passiflora edulis) under fusarium kyushuense and drought stress conditions." Frontiers in Plant Science 13 (2022): 872263
Yang et al. "Genome-wide identification and comprehensive analyses of NAC transcription factor gene family and expression analysis under Fusarium kyushuense and drought stress conditions in Passiflora edulis." Frontiers in Plant Science 13 (2022): 972734.
Rizwan, et al. "Comprehensive genome-wide identification and expression profiling of eceriferum (CER) gene family in passion fruit (Passiflora edulis) under fusarium kyushuense and drought stress conditions." Frontiers in Plant Science 13 (2022): 898307.

In the following manuscript, the authors performed only basic bioinformatics analysis and expression analysis, etc, however, it is strongly recommended to perform, Domain figure, cis-regulatory element analysis, miRNA prediction, Ka/Ks, synteny analysis, collinearity analysis with other genomes, 3D modeling, TFs network analysis, etc, for reference can follow the above mentioned latest publications on passion fruit.
English writing and grammar could be further improved.

---

## Round 0.2 · Major Revisions

Dear author please revise the article as one reviewer has given major revsion.

Reviewer 1 ·

Basic reporting

The revised manuscript is substantially improved and found all the suggestions were incorporated in the revised manuscript.

Experimental design

All the experimental parts were well wriiten.

Validity of the findings

The revised manuscript is found scientifically sound.

·

Basic reporting

no comment'

Experimental design

no comment'

Validity of the findings

The author xie et al., submitted the following manuscript entitled “Genome-wide identification and expression analysis of C2H2 zinc finger proteins in Chinese jujube (Ziziphus jujuba Mill.) in different fruit development stages and under different levels of water stress” after the 1st revision. However, the author has not fully revised the manuscript according to the suggestions and recommendations (comments-1), therefore, It is suggested to address the revisions by mentioning in the answer section the Line numbers for easy understanding.

Additional comments

The author xie et al., submitted the following manuscript entitled “Genome-wide identification and expression analysis of C2H2 zinc finger proteins in Chinese jujube (Ziziphus jujuba Mill.) in different fruit development stages and under different levels of water stress” after the 1st revision. However, the author has not fully revised the manuscript according to the suggestions and recommendations (comments-1), therefore, It is suggested to address the revisions by mentioning in the answer section the Line numbers for easy understanding.

---

## Round 0.3 · accepted · Accept

The reviewers has accepted your revised article. I am happy to Accept it.

Reviewer 1 ·

Basic reporting

The revised manuscript is well written and found satisfactory.

Experimental design

They are appropriate for the manuscript.

Validity of the findings

Based on the present investigation, the present finding is well justified.